

# Impact of Hurricanes Irma and Maria on the PTWC Tsunami Warning Capability for the Caribbean Region

Victor Sardina[1], David Walsh[1], Kanoa Koyanagi[1], Stuart Weinstein[1], Nathan Becker[1],
Charles McCreery[1], and Christa von Hillebrandt-Andrade[2]

[1]Pacific Tsunami Warning Center, NOAA, NWS, Honolulu, Hawaii
[2]US NWS Caribbean Tsunami Warning Program, Mayaguez, Puerto Rico

**Correspondence:** Victor Sardina (victor.sardina@noaa.gov)

**Abstract.** In September 2017, hurricanes Irma and Maria wreaked havoc across the Caribbean region. While obliterating the infrastructure in the Caribbean nations found along their path, both hurricanes gradually destroyed the existing seismic networks. We quantified the impact of the hurricanes on the PTWC tsunami warning capability for the Caribbean region relying on the computation of theoretical earthquake detection and response times after accounting for hurricane-related station outages. The results show that the hurricanes rendered inoperative 38% of the 146 stations available in the Caribbean. Within the eastern Caribbean region monitored by PTWC the hurricanes exacerbated outages to an astonishing 82% of the available 76 seismic stations. Puerto Rico, the Virgin Islands, and the Lesser Antilles suffered the brunt of both hurricanes, and their seismic networks nearly disappeared. The double punch delivered by two successive category 5 hurricanes added up to 02:43 and 04:33 minutes to the earthquake detection and response times, effectively knocking out PTWC's local tsunami warning capabilities in the region. Emergency adjustments, including the temporary reduction of the number of stations required for earthquake detection and ML magnitude release, enabled a faster response to earthquakes in the region than otherwise possible in the aftermath of hurricanes Irma and Maria.

## 1 Introduction

Two category 5 hurricanes on the Saffir-Simpson scale wreaked havoc across the Caribbean region in September 2017. Tropical storm Irma appeared near Cape Verde, off the coast of West Africa on August 30, 2017. By September 6, 2017, Irma had already intensified to a category 5 hurricane, with maximum sustained winds peaking at 295 km/h. It continued moving westward as a category 5 hurricane causing catastrophic damage in Barbuda, Saint Barthélemy, Saint Martin, Anguilla, and the Virgin Islands. On September 10, Irma veered north, towards the strait of Florida, after battering the northern coast of eastern and central Cuba, still as a Category 5 hurricane. Less than a week later, on September 16, tropical storm Maria developed to the east of the Lesser Antilles. By September 18, in less than two days, Maria had intensified to a category 5 hurricane right before making landfall in Dominica, where it obliterated everything in its path. Hurricane Maria then continued moving to the northwest until it made landfall in Puerto Rico as a high-end category 4 hurricane on September 20. Its path along Puerto Rico followed a diagonal trajectory that inflicted catastrophic damage to an infrastructure already left in tatters after hurricane Irma just two weeks prior.





Meanwhile, in Hawaii, the duty scientists at the PTWC witnessed how most of the seismic data streams coming in from the eastern Caribbean gradually disappeared in a matter of weeks. These observations alone, however, do not sufficiently elucidate the impact of both hurricanes on tsunami warning operations. We quantify the impact of both hurricanes on the PTWC tsunami warning capabilities for the Caribbean by first computing the theoretical detection time of the first arriving P and S seismic waves across the region. We then manipulate these computations to generate maps illustrating the spatial distribution of the additional earthquake detection and response time delays in seconds triggered by both hurricanes.

## 2 Computational Methodology

To compute the detection times of the first arriving P and S waves we rely on the procedure applied by *Sardina et al.* (2018a). We first apply the Tau-P method (*Buland and Chapman*, 1983) using the AK135 earth model (*Kennett et al.*, 1995) via the TauP software package (*Crotwell et al.*, 1999) and generate travel time tables for the first arriving P and S waves. We then use these tables as input to a multithreaded C++ application written to generate 2D grids for different combinations of receiver stations, azimuth gap constraints, and lookup regions. In all computations we place the hypocenter at 10 km, the depth reported as the most frequent for Caribbean earthquakes processed by the PTWC (*Sardina et al.*, 2018b).

## 3 P and S Wave Detection Times in the Caribbean

When monitoring the seismic activity within a given area geoscientists traditionally grade preliminary hypocenter determinations with smaller azimuthal gaps as having better overall quality than those with larger ones. Given both the topology of the seismic network in the Caribbean shown in Figure 1, and its inhomogeneous density of stations, however, it turns much slower, and therefore quite impractical for tsunami warning purposes, to set a maximum azimuthal gap restriction before releasing preliminary earthquake detections. Consequently, to locate earthquakes worldwide the PTWC teleseismic picker relies on detection of the P waves at a minimum of five stations regardless of the azimuthal gap, while the associator releases a preliminary location for further processing once it has successfully associated at least eight P-picks regardless of the azimuthal gap. We mimicked the settings of the real-time monitoring system for the Caribbean by computing the detection time of the first arriving P-wave at a minimum of 8 stations, without azimuthal gap restrictions, after assuming that all stations provide data with minimal latency. Analysis of the distribution of the computed detection times shown in Figure 1a reveals that by relying on detection of the first arriving P-wave at the 8 closest seismic stations we could detect seismic events in the Caribbean within 90 and 60 seconds of origin time in 83% and 47% of the region. Moreover, in areas covered by a denser seismic network, such as Puerto Rico and the Lesser Antilles, we could detect earthquakes within 30 seconds of origin in 6.9% of the region, which represents 14.7% of the 60 second detection area.

The histogram on the upper left corner of Figure 1b illustrates the distribution of the median latency for the seismic data streams ingested into the PTWC system. We attained these median latency values from the analysis of 628 latency log files covering the second half of 2017. We can then treat these median values as representative of the most common network status





and use them as a baseline to quantify deviations from the norm. As shown in the histogram, station outages usually render useless 23% (34) of the 146 stations, while another 52% have data latencies longer than 10 seconds. We can expect only a quarter (25%) of the stations to provide data with latencies under 10 seconds. The contour lines in Figure 1b illustrate the spatial distribution of the detection time of the first arriving P wave at a minimum of 8 stations once we take into account

these median data latencies and station outages. Comparison of Figure 1b to Figure 1a reveals that the 90, 60, and 30 second detection areas shrink from 83% to 54%, 47% to 14%, and 6.9% to 0.25% of the region, thereby undergoing a 35%, 70%, and 96% reduction in area.

Likewise, Figure 2 illustrates the spatial distribution of the detection time of the first arriving S-wave at a minimum of 5 stations, without azimuthal gap restrictions, after assuming that all stations have no data latency issues. As observed in Figure

2a, we could detect the S waves at the five closest stations within 90, 60, and 30 seconds from origin time across 47%, 18%, and 2.7% of the total area. Once we take into account the data latencies and station outages, however, inspection of Figure 2b reveals that the 90, 60, and 30 second detection areas shrink from 47% to 23%, 18% to 6%, and 2.7% to 0.2% of the region, equivalent to a 51%, 66%, and 92% reduction in area.

We further illustrate the impact of the data latencies and station outages in Figure 3, attained by a) subtracting the P-wave

detection times shown in Figure 1a from those in Figure 1b, and b) subtracting the S-wave detection times shown in Figure 2a from those in Figure 2b. As illustrated in Figure 3a,b we can expect station outages and data latencies to cause detection delays of at least 15 seconds throughout at least 85% of the Caribbean. We can also expect P and S wave detection delays of more than 30 seconds to affect 28% and 34% of the region, respectively. Longer detection delays affect heavily the northwest quadrant, including Cuba, the Cayman Islands, Jamaica, Haiti, and the Dominican Republic, due both to a more sparse seismic

network to the northwest, and relatively frequent station outages and longer data latencies.

Relying on detection of the P waves at the closest seismic stations without setting a maximum azimuth gap requirement results in faster earthquake detections, but also in larger azimuthal gaps. As illustrated in Figure 4a, after taking into account the median station outages we can expect azimuthal gaps larger than 180°in 59.5% of the Caribbean region, and across 74% of the rectangular area monitored by the PTWC, shown in more detail in Figure 4b. When detecting earthquakes along the trench

axes, both to the north and south of Puerto Rico, we can expect azimuthal gaps of more than 270°degrees. These theoretical azimuthal gaps match the actual data, as for instance, during 2017 the PTWC hypocenter determinations for earthquakes in the Caribbean had azimuthal gaps characterized by a median of 180.3 °. Despite rather large azimuthal gaps, however, the PTWC epicenter offsets had a median value of 14.3 km.

## 4   P and S Wave Detection Times in the Caribbean after Hurricanes Irma and Maria

Hurricane Irma devastated the region's infrastructure, and this in turn contributed considerably to magnify the catastrophic damage found along the path of hurricane Maria less than two weeks later. The histogram in Figure 5a illustrates the status of the Caribbean seismic network after hurricane Irma on September 10, 2017. Data outages at 53 stations now account for 36% of all 146 stations. Comparison of Figure 5a to Figure 1b reveals that the 19 additional station outages attributed to Irma further



reduced the 90, 60, and 30 second P-wave detection areas to 47%, 11.5% and 0.16% of the region, respectively. Likewise, comparison of Figure 5b to Figure 2b reveals that after hurricane Irma the 90, 60, and 30 second S-wave detection areas shrunk to just 19.3%, 4.8%, and 0.1% of the total area, respectively.

The status of the seismic monitoring for tsunami warning purposes in the Caribbean deteriorated even further less than two

5   weeks later with the arrival of hurricane Maria. The histogram in Figure 6 shows that after hurricane Maria, on September 23, 2017, the PTWC had lost access to 90 (62%) of the 146 seismic stations available in the Caribbean, with an overwhelming number of them located in the vicinity of Puerto Rico, the Virgin Islands, and the Lesser Antilles. The paths of both hurricanes, derived from the advisories issued by the National Hurricane Center, and plotted as two white vortex tracks in all pertinent figures, match the location of the station outages. Moreover, the concentration of black triangles representing station outages

in Figure 6, for instance, underline the fact that hurricanes Irma and Maria caused an unprecedented, massive blackout of the seismic networks in the eastern Caribbean.

As illustrated in Figure 6a, after hurricane Maria, the 90 (yellow) and 60 (orange) second P-wave detection areas now cover just 24.3% and 1.35% of the region, respectively. Likewise, Figure 6b shows that after hurricane Maria the 90 and 60 second S-wave detection areas shrunk to just 8.5% and 0.9% of the total area, while the 30 second detection area completely disappears

for both the P and S wave detection.

We further highlight how the eastern Caribbean region suffered the brunt of both hurricanes in Figures 7 and 8, attained by a) subtracting the median P-wave detection times shown in Figure 1b from those after hurricane Maria shown in Figure 5b, and b) subtracting the median S-wave detection times shown in Figure 2b from those after hurricane Maria shown in Figure 6b. This essentially isolates the effect of both hurricanes on the detection times, thereby allowing to quantify their impact as

additional, hurricane-related detection delays in seconds.

As observed in Figure 7a, when compared to normal operational conditions shown in Figure 1b, hurricanes Irma and Maria caused additional P-wave detection delays of more than 60 seconds (01:00) across 19% of the Caribbean region. The longer delays, however, appear heavily concentrated within the rectangular area monitored by the PTWC in the eastern Caribbean, shown in greater detail in Figure 7b. The combined effect of both hurricanes left just 13 stations available for seismic monitor-

ing, with only 3 of them located in the Lesser Antilles. This in turn results in additional P-wave detection delays of 60∼163 seconds (01:00∼02:43) across 51% of the eastern Caribbean, including Puerto Rico, the Virgin Islands, and the Lesser Antilles.

Similarly, as illustrated in Figure 8a, when compared to the normal operational conditions illustrated in Figure 2b, the hurricanes caused additional S-wave detection delays of 60 seconds (01:00) or more across 23% of the Caribbean region. Figure 8b corroborates how the longer hurricane-triggered delays concentrate heavily within the eastern Caribbean, where

additional S-wave detection delays of 60∼273 seconds (01:00∼04:33) affect 60% of the area.

## 5   Mitigation of the Impact of Hurricanes Irma and Maria on P and S Wave Detection Times

The chronic P and S wave detection delays triggered by the station outages triggered by hurricanes Irma and Maria, illustrated in Figure 6, made the 8 P-phase and 5 S-phase criteria unsuitable for local tsunami warning operations. The unprecedented



earthquake detection delays shown in Figure 7 and 8 prompted emergency adjustments to the PTWC local processing system for the eastern Caribbean. As illustrated in Figure 9, these adjustments consisted in a) reducing to 4 the number of P-phase picks required for event detection, and b) reducing to 2 the number of S-phase picks required for preliminary $M_L$ magnitude computation. While these steps improve detection times, they also result in less stable hypocenters and magnitude estimates, necessitating additional review by the duty geoscientists.

Relying on detection of the first arriving P wave at 4 stations instead of 8, however, results in a significant reduction of the impact of the hurricanes on the P-wave detection times, from a maximum additional delay of 163 seconds (02:43) shown in Figure 7b to the 66 seconds (01:06) shown in Figure 9a. Moreover, the area affected by additional P-wave detection delays of more than 60 seconds shrinks from 51% in Figure 7b to 2.7% of the whole area in Figure 9a, equivalent to a 95% reduction in area.

Likewise, relying on detection of the first arriving S wave at 2 stations instead of 5 for preliminary $M_L$ computation results in a visible reduction of the impact of the hurricanes on the S-wave detection times, from a maximum additional delay of 273 seconds (04:33) shown in Figure 8b to the 120 seconds (02:00) shown in Figure 9b. Moreover, the area affected by S-wave detection delays of more than 60 seconds shrinks from 60% in Figure 8b to 30% of the whole area in Figure 9b, equivalent to a 50% reduction in area.

## 6 Response Times in the Aftermath of Hurricanes Irma and Maria

The results discussed so far underscore both the how and why hurricanes Irma and Maria had the worst repercussions on tsunami warning operations in the eastern Caribbean. Consequently, when discussing their impact on the response times we will focus on the rectangular area monitored by the PTWC local monitoring system for Puerto Rico, the Virgin Islands, and the Lesser Antilles shown in Figure 8. To illustrate the spatial distribution of the theoretical response times in the wake of both hurricanes we converted the S-wave detection maps reflecting the hurricanes' impact into theoretical response time maps by adding 110 seconds to account for a) the 30 second S-wave coda window required for $M_L$ magnitude computation, and b) the historical median of 80 seconds needed to review the available data and compose a message after *Sardina et al.* (2018b).

We computed the theoretical response times for detection of the first arriving S wave at a minimum of 5 stations by applying the operation: $Figure\,10a = (Figure\,6b + 110)$. As indicated by the contour lines in Figure 10a, relying on S-wave detection at a minimum of 5 stations leads to response times of $240\sim463$ seconds ($04{:}00\sim07{:}43$) for any earthquake located to the east of the Dominican Republic. The diameter of the circles plotted in Figure 10 indicates the magnitude of 12 local earthquakes processed by the PTWC between September 12, 2017 and January 7, 2018. Their color reflects the theoretical response time read directly from the map.

To compute the theoretical response times after detection of the first arriving S wave at a minimum of 2 stations we applied the operation: $Figure\,10b = (Figure\,2b + Figure\,8b + 110)$. The color assigned to the 12 earthquake symbols in Figure 10b now indicates the actual PTWC response times using the same time scale applied to the contours. As we can observe, the color assigned to all but one earthquake differs from the color of the contour bands underneath by no more than the equivalent of





$\pm 30$ seconds. We can attribute these differences to a) faster or slower manual review and message composition than the median 80 seconds, and b) the gradual repair and availability of more seismic stations as part of the hurricanes' recovery process. This corroborates that adjusting the settings of the monitoring system to rely on detection at 4 stations, plus the computation of preliminary $M_L$ magnitudes at 2 stations reduced considerably the additional delays attributed to the impact of the hurricanes.

## 7  Conclusions

We assessed the devastating impact of hurricanes Irma and Maria on the Pacific Tsunami Warning Center's (PTWC) tsunami warning capabilities for the Caribbean relying on the computation of theoretical earthquake detection and response times. In these computations we accounted for the topology of the seismic network, but also the median data latencies and the additional station outages attributed to both hurricanes in September of 2017. Analysis of the results allows us to draw the following conclusions:

- Analysis of the log files documenting the latency of the seismic data streams ingested into the PTWC system from the Caribbean during the second half of 2017 reveals that under normal operational conditions we can expect a) outages at 23% (34) of the 146 stations, b) data latencies exceeding 10 seconds for another 52%, and c) just a quarter (25%) of all data streams with latencies under 10 seconds.

- Theoretical computation of the detection time of the first arriving P wave in the Caribbean region at a minimum of 8 stations reveals that under normal operational conditions we can expect data latencies and station outages to cause P-wave detection delays exceeding 15 seconds across 85% of the Caribbean, with delays of 30∼59 seconds (00:30∼00:59) affecting 28% of the region (Figure 1 and Figure 3a).

- Theoretical computation of the detection time of the first arriving S wave in the Caribbean region at a minimum of 5 stations reveals that we can expect data latencies and station outages to cause S-wave detection delays of 15 seconds or more across 86% of the Caribbean, with detection delays of 30∼92 seconds (00:30∼01:32) affecting 34% of the region (Figure 2 and Figure 3b).

- Relying on detection of the first arriving P wave in the Caribbean region at a minimum of 8 stations results in preliminary earthquake locations with azimuthal gaps of more than $180°$ degrees across 59.5% of the Caribbean region (Figure 4a), and 76% of the rectangular area monitored by the PTWC in the eastern Caribbean (Figure 4b).

- After hurricane Irma, on September 10, 2017, the PTWC had lost access to 36% (53) of the 146 stations available in the Caribbean. The 19 station outages attributed to hurricane Irma caused additional P and S wave detection delays that reduced the 90, 60, and 30 second detection areas to a) 47%, 11.5%, and 0.16% of the region for P-wave detection, and b) 19.3%, 4.8%, and 0.1% of the region for S-wave detection (Figure 5).

- After hurricane Maria, on September 23, 2017, the PTWC had lost access to 62% (90) of the 146 stations available in the Caribbean. These unprecedented, massive seismic station outages attributed to hurricanes Irma and Maria resulted



in additional P and S wave detection delays that reduced the 90 and 60 second detection areas to a) 24.3% and 1.35% of the region for P-wave detection, and b) 8.5% and 0.9% of the total area for S-wave detection, while the 30 seconds detection area completely disappeared (Figure 6).

- The hurricanes caused additional P and S wave detection delays of more than 15 seconds across 43% and 45% of the Caribbean region (Figure 7a and Figure 8a), respectively. The longest detection delays, however, concentrate heavily along the path of both hurricanes in the eastern Caribbean, where additional P and S wave detection delays exceeding 15 seconds affect 88% and 78% of the area, respectively. Moreover, within the rectangular area monitored by the PTWC P-wave detection delays of 60∼163 seconds (01:00∼02:43) affect 51% of the area (Figure 7b), while S-wave detection delays of 60∼273 seconds (01:00∼04:33) affect 61% of the area (Figure 8b).

- Computation of the theoretical response times for the eastern Caribbean while accounting for the impact of hurricanes Irma and Maria results in response times of 240∼463 seconds (04:00∼07:43) for any earthquake located to the east of the Dominican Republic (Figure 10a). The theoretical response times based on the detection of the S waves at a minimum of 2 stations, however, show good agreement with the actual PTWC response times for 12 events processed between September 12, 2017, and January 7, 2018, within ±30 seconds (Figure 10b). This corroborates that adjusting the monitoring system to rely on detection of the first arriving P wave at 4 stations instead of 8 for event location, plus detection of the first arriving S wave at 2 instead of 5 stations to compute preliminary $M_L$ magnitudes reduced considerably the additional message delays attributed to the impact of the hurricanes (Figure 10b).

- Theoretical computation and analysis of the impact of the additional station outages attributed to hurricanes Irma and Maria on the detection and response times in the Caribbean reveals that after hurricane Maria the PTWC no longer had a local tsunami warning capability for Puerto Rico and the Virgin Islands. The massive blackout of seismic stations in the eastern Caribbean made it operationally impractical to either detect and locate local earthquakes, or compute $M_L$ magnitudes as low as 3.8 in a timely manner. Notwithstanding, the PTWC still maintained a regional tsunami warning capability for the Caribbean, including Puerto Rico and the Virgin Islands, relying on its teleseismic monitoring system for the region, albeit for magnitude 6.0 or larger magnitude earthquakes, and response times quite likely to exceed 6 minutes for events located in the eastern Caribbean.

- The devastating impact of hurricanes Irma and Maria on the PTWC local tsunami warning capabilities for Puerto Rico and the Virgin Islands highlights the vital, and potentially life saving role of educating the population to self-evacuate in the event of prolonged or strong ground shaking instead of waiting for official tsunami messages.

- When reinstalling damaged stations and rebuilding the supporting infrastructure network operators should consider to hurricane-proof at least a subset of their seismic stations, so as to maintain a minimum earthquake monitoring and local tsunami warning capability even if impacted by category 5 hurricanes such as Irma and Maria. To facilitate station selection, a compilation of usage statistics combined with the generation of theoretical detection and response time maps as done in this study should reveal the stations most valuable to any regional monitoring network.



*Competing interests.* The authors declare that they have no conflict of interest.

*Acknowledgements.* In this study we used the $TauP$ sofware package (*Crotwell et al.*, 1999) to generate travel time tables for the first
arriving P and S waves. We also used the Qt C++ Framework (https://www.qt.io) to further develop the multithreaded C++ application used
to generate the 2D travel time grids subsequently manipulated and plotted via the Generic Mapping Tools (GMT) (*Wessel et al.*, 2013)
5   software package. We retrieve the hurricane path data from the National Hurricane Center (NHC), 2017 Tropical Cyclone Advisory Archive
onlite at: https://www.nhc.noaa.gov/archive/2017.





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


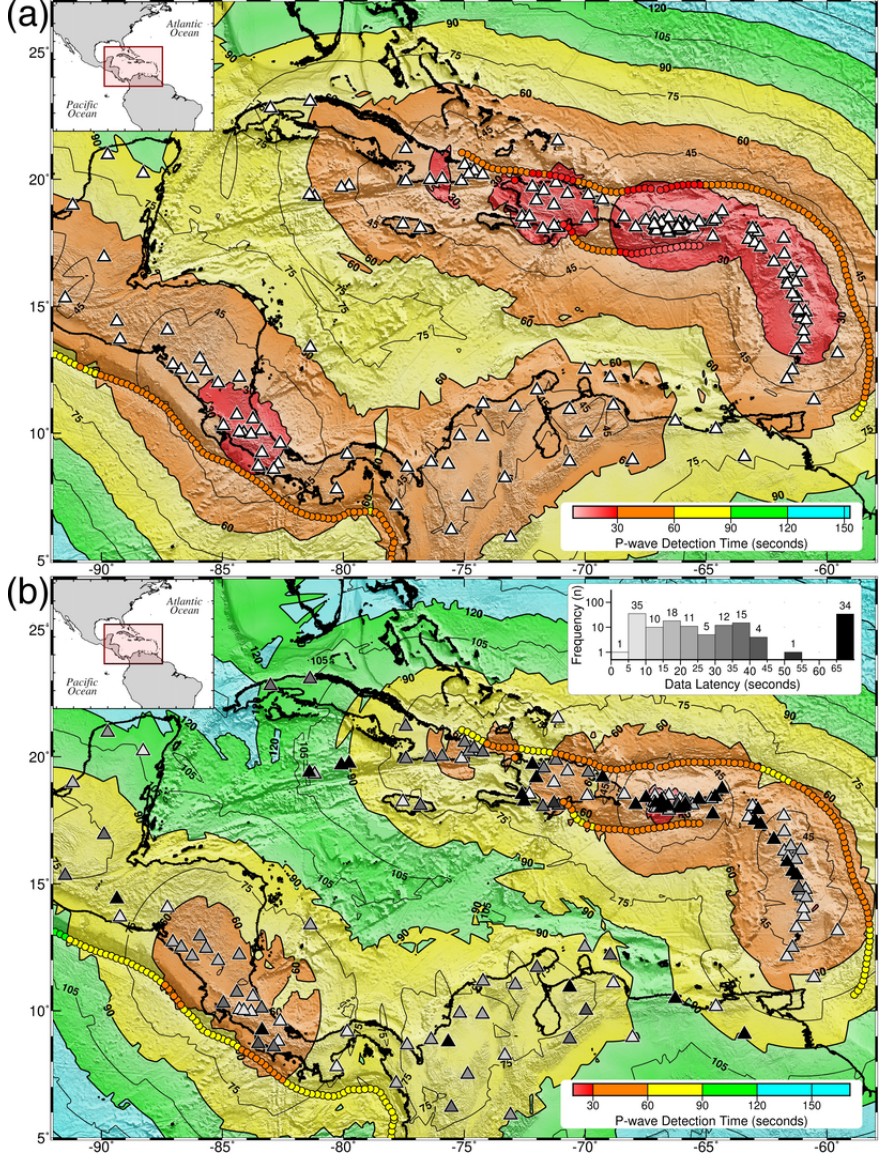

**Figure 1.** Theoretical detection time of the first arriving P wave within the Caribbean region at a minimum of 8 stations, regardless of azimuthal gap after a) assuming that all seismic stations (white triangles) contribute data with no significant latencies, and b) taking into account the median data latencies and station outages. Station outages at 23% (34 black triangles) of the 146 seismic stations, and data latencies longer than 10 seconds for another 52% leave only 25% of the network with latencies under 10 seconds. Consequently, the 90 (yellow), 60 (orange) and 30 (red) second detection areas shrink from 83% to 54%, 47% to 14%, and 6.9% to 0.25% of the region, thereby undergoing a 35%, 70%, and 96% reduction in area, respectively. The small, contiguous circles show the computed P-wave detection times sampled every 25 km along the trench axes.



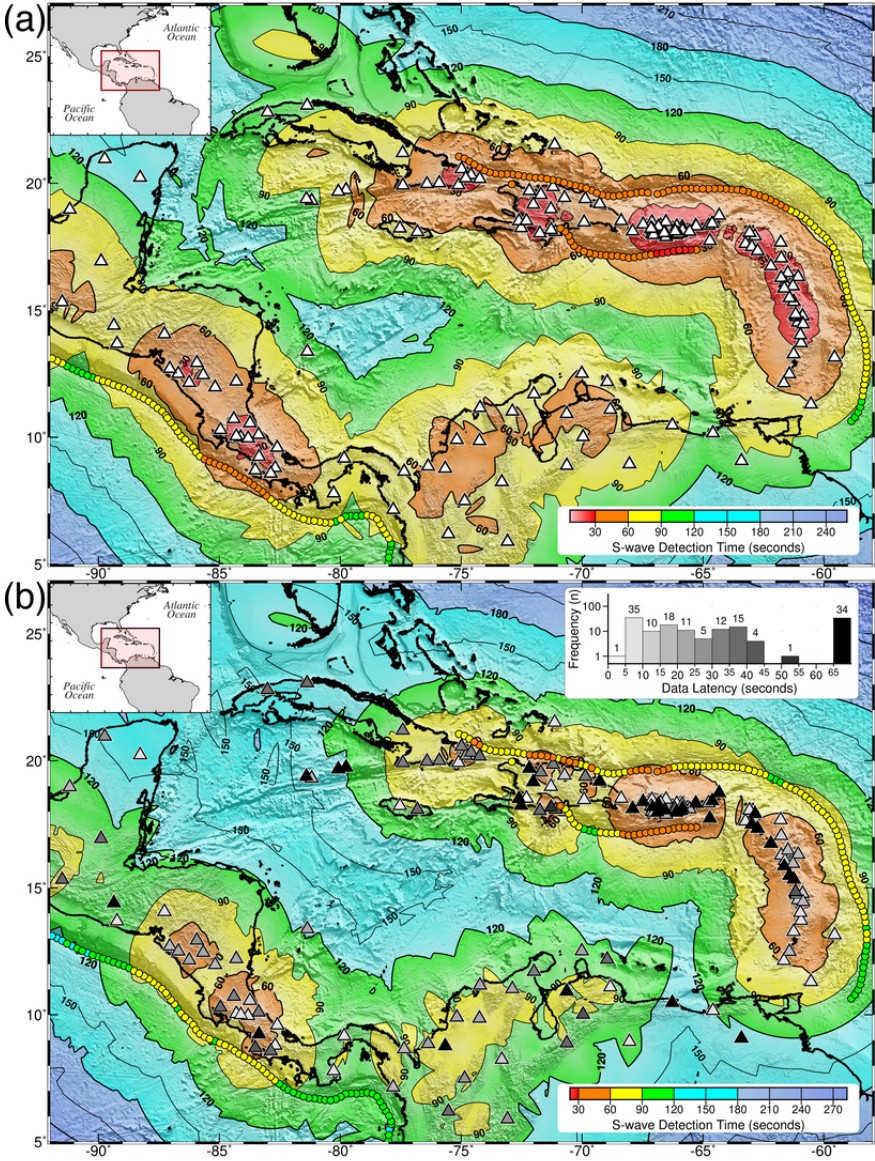

**Figure 2.** Theoretical detection time of the first arriving S wave within the Caribbean region at a minimum of 5 stations, regardless of azimuthal gap after a) assuming that all seismic stations (white triangles) contribute data with no significant latencies, and b) taking into account the median data latencies and station outages. Station outages at 23% (34 black triangles) of the 146 seismic stations, and data latencies longer than 10 seconds for another 52% leave only 25% of the network with latencies under 10 seconds. Consequently, the 90 (yellow), 60 (orange), and 30 (red) second detection areas shrink from 47% to 23%, 18% to 6%, and 2.7% to 0.2% of the region, equivalent to a 51%, 66%, and 92% reduction in area, respectively. The small, contiguous circles show the computed S-wave detection times sampled every 25 km along the trench axes.



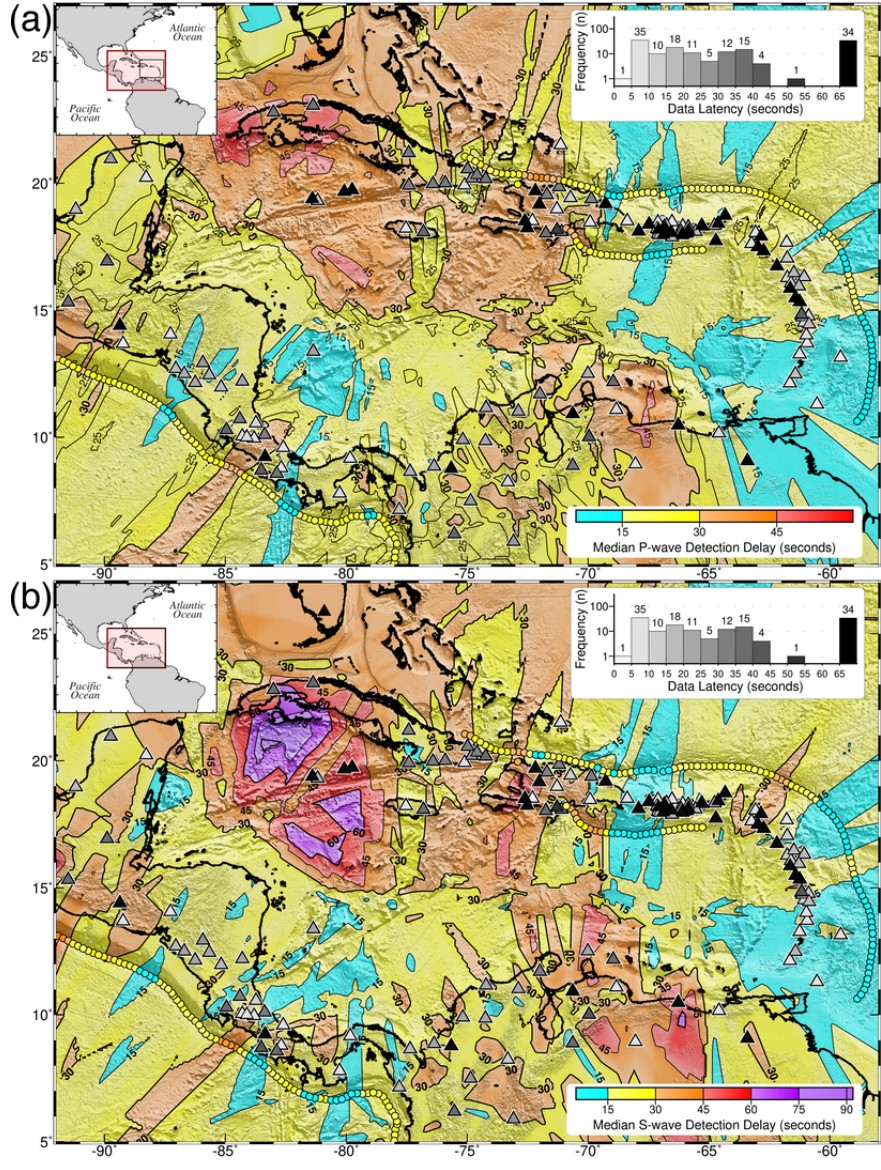

**Figure 3.** Detection delay in seconds once we account for the data latencies and station outages on the detection time of the first arriving P and S waves after a) subtracting the P-wave detection times shown in Figure 1a from those in Figure 1b, and b) subtracting the S-wave detection times shown in Figure 2a from those in Figure 2b. We can expect outages at 23% (black triangles) and data latencies longer than 10 seconds at 52% of the 146 stations to cause detection delays of at least 15 seconds across 85% of the region. Likewise, we can expect detection delays of more than 30 seconds to affect 28% and 34% of the region in (a) and (b) respectively. The small, contiguous circles show the computed detection delays sampled every 25 km along the trench axes.



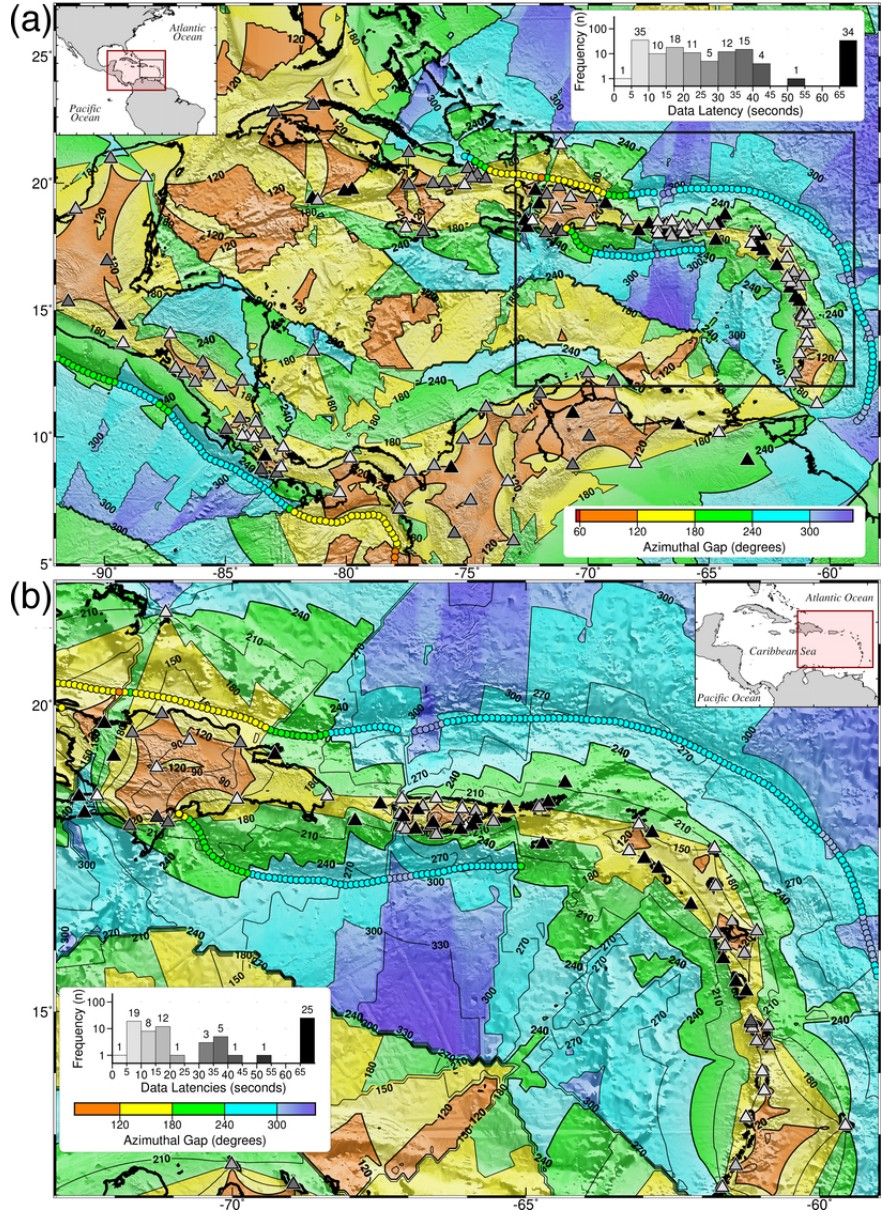

**Figure 4.** Azimuthal gap in degrees resulting from the detection of the first arriving P wave within the Caribbean at a minimum of 8 stations, without azimuth gap restriction, after taking into consideration the median data latencies and station outages within a) the Caribbean region, and b) within the rectangular area monitored by the PTWC local processing system for the eastern Caribbean. Under normal operational conditions we can expect azimuthal gaps under $180°$(yellow) and $120°$(orange) in just 40.5% and 10.9% of the Caribbean region, respectively. Within the eastern Caribbean area shown in b) we can expect azimuthal gaps under $180°$ and $120°$ in just 26% and 4.5% of the total area, mostly within smaller sections located along the axis of the most densely instrumented areas.





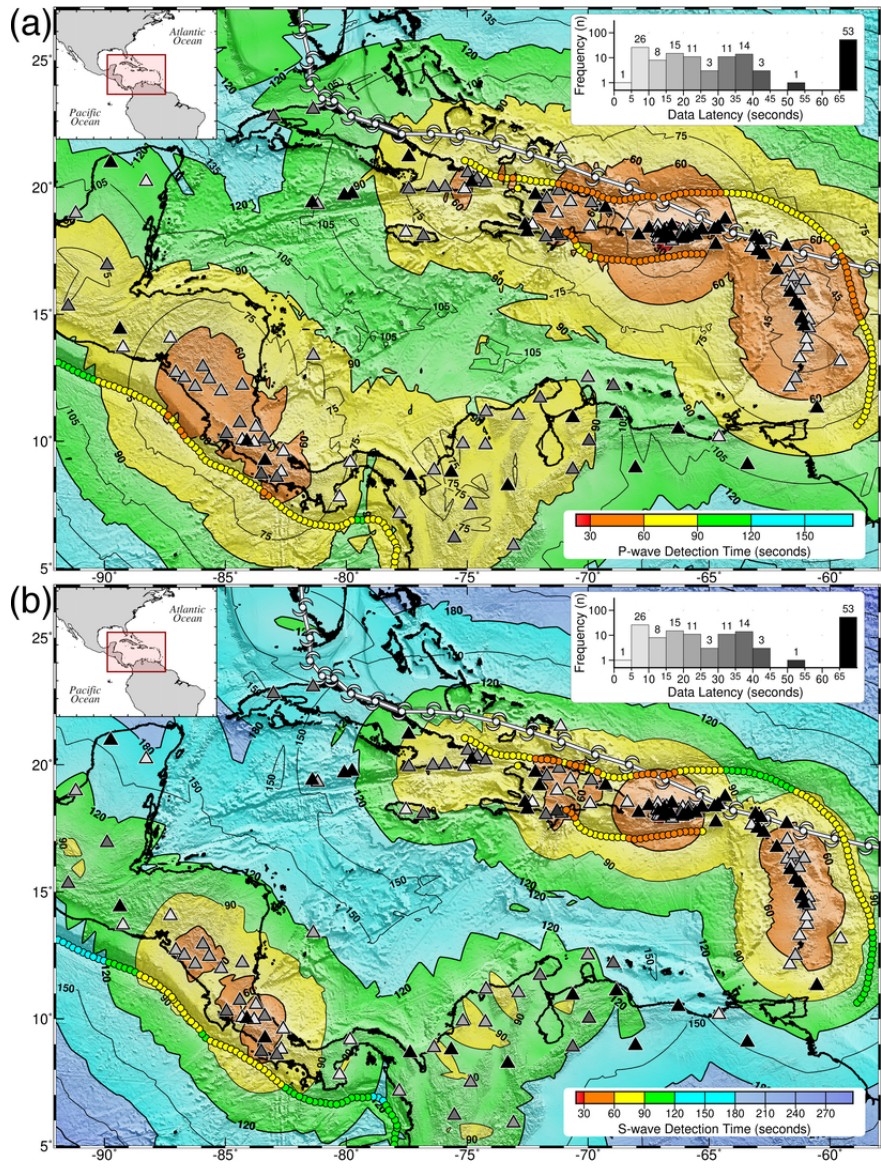

**Figure 5.** Theoretical detection time of the first arriving P and S waves in the Caribbean after hurricane Irma on September 10, 2017. The 19 additional station outages attributed to Hurricane Irma now add up to 36% (53 black triangles) of the 146 stations. These outages in turn reduced the 90 (yellow), 60 (orange), and 30 (red) detection areas to a) 47%, 11.5%, and 0.16% of the total area for P-wave detection, and b) 19.3%, 4.8%, and 0.1% of the region for S-wave detection. The small, contiguous circles show the computed detection times sampled every 25 km along the trench axes. White vortex tracks show the path of hurricane Irma.





**Figure 6.** Theoretical detection time of the first arriving P and S waves in the Caribbean after hurricane Maria on September 23, 2017. The 37 additional station outages attributed to Hurricane Maria now add up to 61% (90 black triangles) of the 146 stations. These unprecedented station outages further reduced the 90 (yellow) and 60 (orange) second detection areas to 24.3% and 1.35% of the region for P-wave detection, and b) 8.5% and 0.9% of the total area for S-wave detection. The 30 seconds detection area disappears in both (a) and (b). The small, contiguous circles show the computed S-wave detection times sampled every 25 km along the trench axes. White vortex tracks show the path of hurricanes Irma to the north and Maria to the south.





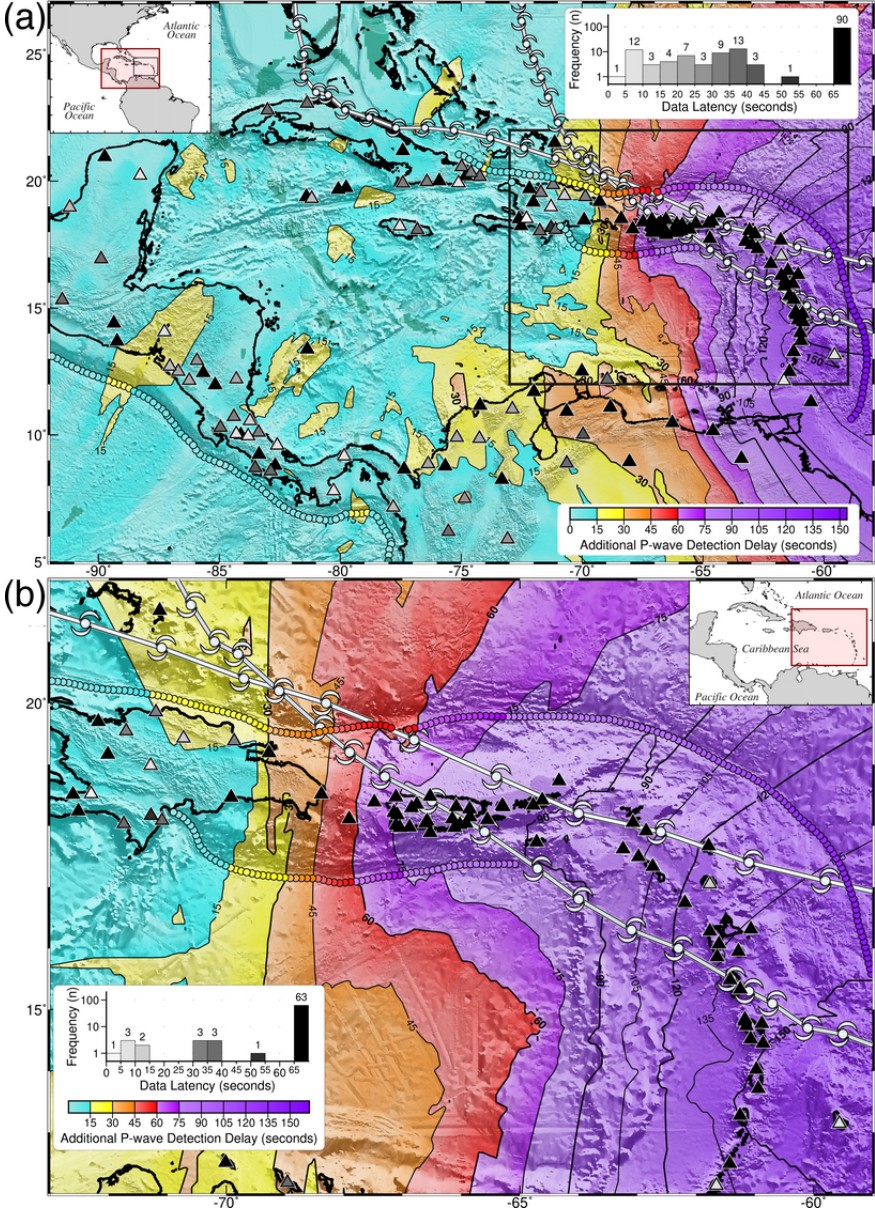

**Figure 7.** Impact of hurricanes Irma and Maria on the detection time of the first arriving P wave at a minimum of 8 stations in the Caribbean after a) subtracting the median P-wave detection times shown in Figure 1b from those in Figure 6a, and b) zooming into the rectangular area monitored by the PTWC local processing system for the Caribbean. The hurricanes caused additional P-wave detection delays of more than 15 seconds across 43% of the Caribbean region. Within the eastern Caribbean shown in (b), however, additional P-wave detection delays of more than 15 seconds affect 88% of the total area, with delays of 60∼163 seconds (01:00∼02:43) affecting 51% of the eastern half of the area. The small, contiguous circles show the computed P-wave detection delays sampled every a) 25 km, and b) 10 km along the trench axes. White vortex tracks show the path of hurricanes Irma to the north and Maria to the south.





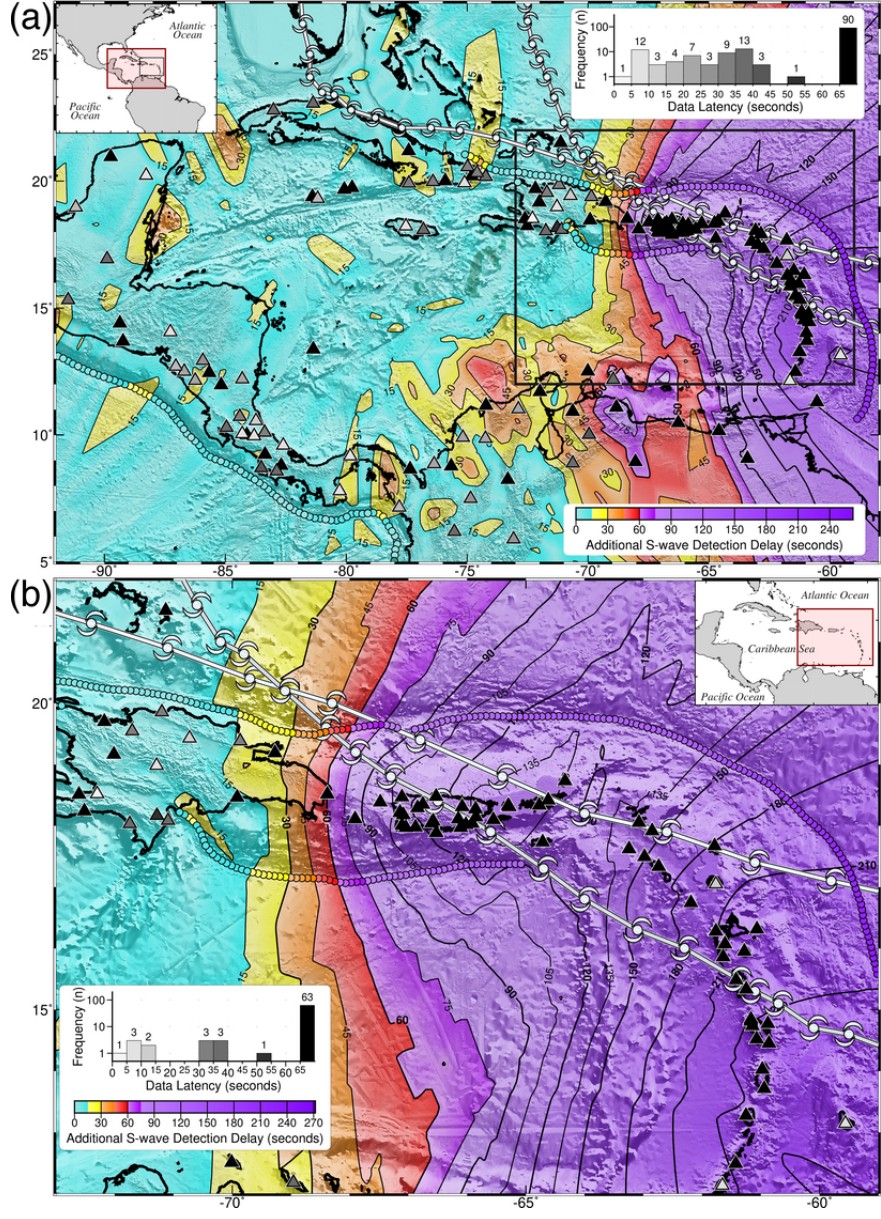

**Figure 8.** Impact of hurricanes Irma and Maria on the detection time of the first arriving S wave at a minimum of 5 stations in the Caribbean after a) subtracting the median S-wave detection times shown in Figure 2b from those in Figure 6b, and b) zooming into the rectangular area monitored by the PTWC local processing system for the Caribbean. The hurricanes caused additional S-wave detection delays of more than 15 seconds across 45% of the Caribbean region. Within the eastern Caribbean shown in (b), however, additional S-wave detection delays of more than 15 seconds affect 78% of the area, with delays of 60~273 seconds (00:15~04:33) affecting 61% of the area. The small, contiguous circles show the computed S-wave detection delays sampled every a) 25 km, and b) 10 km along the trench axes. White vortex tracks show the path of hurricanes Irma to the north and Maria to the south.



Natural Hazards
and Earth System
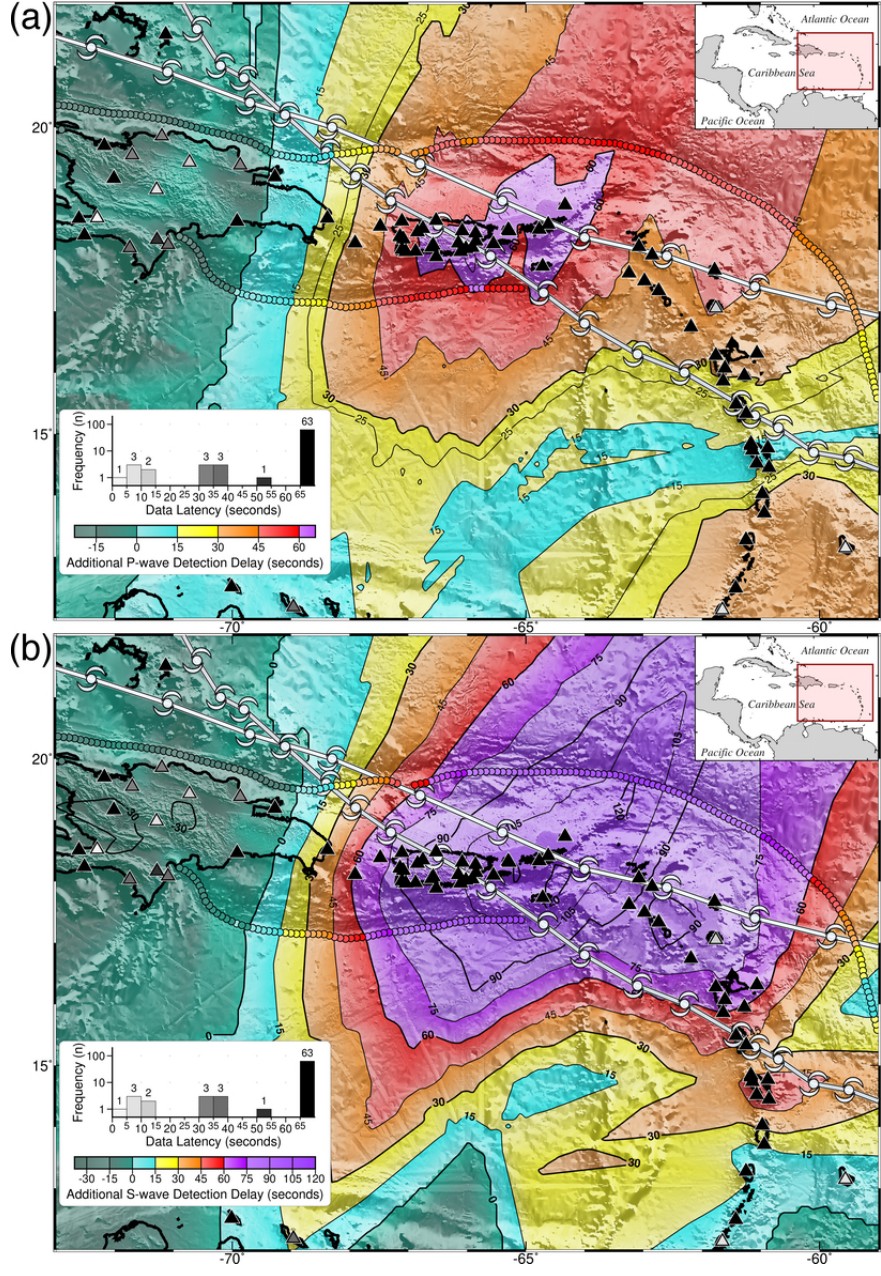

**Figure 9.** Mitigation of the impact of hurricanes Irma and Maria on the detection time of the first arriving P and S waves by a) reducing to 4 stations the number of P-picks required for event detection, so that we have additional P-wave detection delays with a maximum of 66 instead of the 163 seconds (02:43) shown in Figure 7b, and b) reducing to 2 stations the number of S-picks required for computation and release of a preliminary $M_L$ magnitude, thereby having additional S-wave detection delays with a maximum of 120 instead of the 273 seconds (04:33) shown in Figure 8b. The small, contiguous circles show the computed P and S wave detection delays sampled every 10 km along the trench axes. White vortex tracks show the path of hurricanes Irma to the north and Maria to the south.




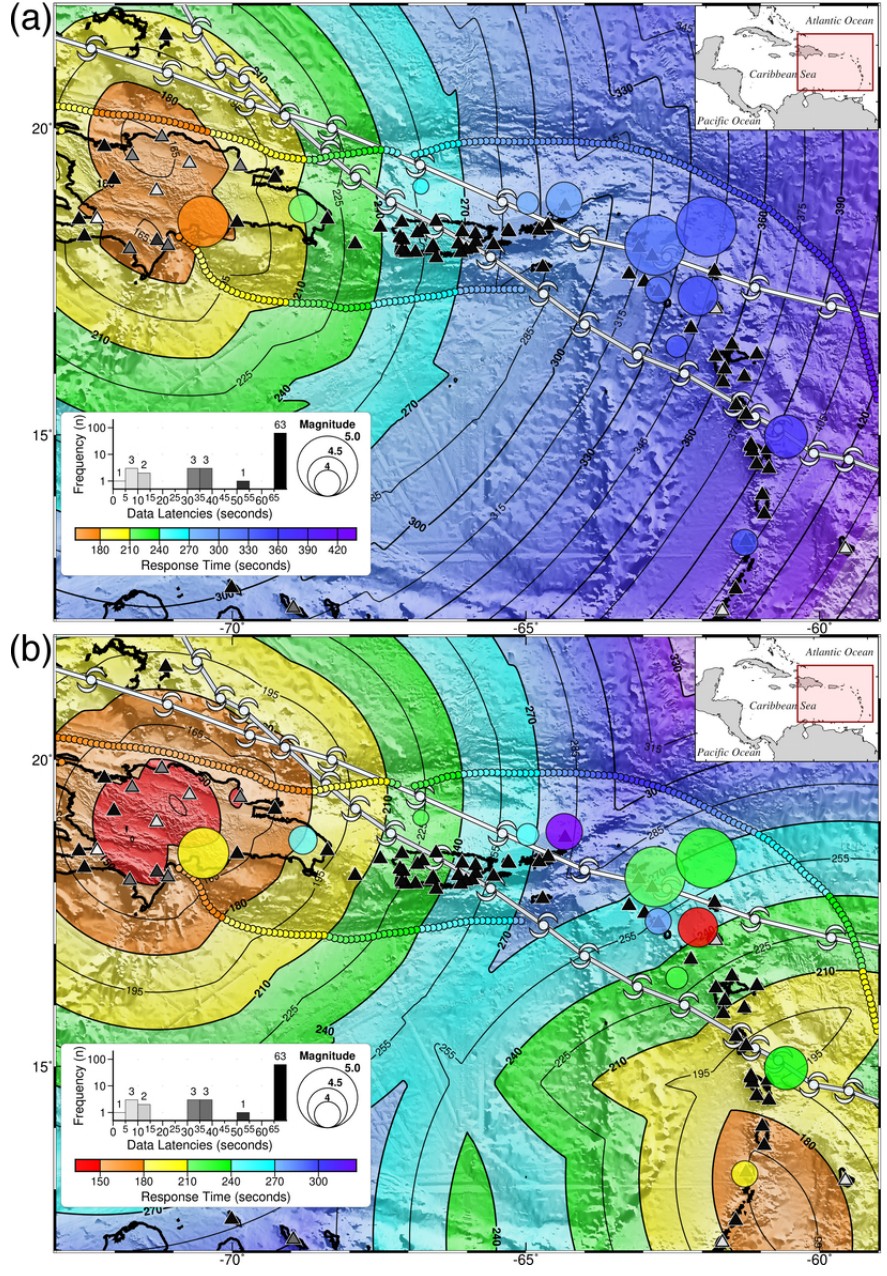

**Figure 10.** Theoretical response times within the eastern Caribbean region monitored by the PTWC after accounting for the impact of hurricanes Irma and Maria. Response times obtained by a) adding 110 seconds to the detection times of the first arriving S wave at the closest 5 stations, namely: $Figure\,10a = (Figure\,6b + 110)$, and b) adding 110 seconds to the detection times of the first arriving S wave at the closest 2 stations applying the operation: $Figure\,10b = (Figure\,2b + Figure\,8b + 110)$. The diameter of the circles indicates the catalog magnitude for 12 local earthquakes processed by PTWC between September 12, 2017 and January 7, 2018, while their color indicates a) the theoretical response times read directly from the map, and b) the actual PTWC response times for the 12 earthquakes. The small, contiguous circles illustrate the computed response times sampled every 10 km along the trench axes. White vortex tracks show the path of hurricanes Irma to the north and Maria to the south.