# Peer review of "Impact of Hurricanes Irma and Maria on the PTWC Tsunami Warning Capability for the Caribbean Region"

_Natural Hazards and Earth System Sciences, 2019_

## Referee Comment (RC1) · Francois Schindele (Referee) · 29 Mar 2019

The authors described the impact of two recent 2017 hurricanes on the availability of seismic data in the region impacted by the hurricanes. This topic is important to demonstrate the robustness or the potential issues of the capacities of a Tsunami warning system after impact of large hurricanes that could occur in basins like Caribbean sea or other basins as the South western-Pacific.

Nevertheless, as it is presented, the results of this study are not complete and a major revision is needed before publication.

[Figure]

The first comment is related to the magnitude referred. As related to fast assessment of earthquakes parameters, it is questionable to mention only the ML computation (page 5 line 11). It is well known that most of the Tsunami warning centers compute Mw, that is the more accurate magnitude in particular for last earthquakes and also typical "tsunami earthquake" events (see Kanamori 1975). Why the authors don't take into account the computation of Mw, and the variations of Mw accuracy depending on the station available after the hurricanes ?

The second comment more general, the authors don't mention the reduction of the accuracy due to lack of data for the fast assessment of seismic parameters (location, depth, magnitude, etc...) To validate the results and conclusions of that study, a complementary study, using data set of recent large earthquakes in the region and eliminate the corresponding data (removing a set of data , i) the set corresponding to the stations stopped in consequence of the two 2017 hurricanes ; ii) other sets of data with several various hypothesis of path of future hurricane, southern, western Caribbean sea. . .) would be the best demonstration, and quantify the impact of such weather disaster on the capacities of tsunami monitoring networks and warning systems.

Another additional point related to fast seismic parameters computation and accuracy. The W-Phase centroid moment tensor computation is used at PTWC to get a fast tsunami threat forecast. The authors should provide the impact of data of large set of stations missing to the accuracy of the results of computation of W-Phase centroid moment tensor, considering only the set of stations available after the hurricanes. Similar complementary study could be performed considering one of the recent large earthquake in the region ( M > 7,0) and eliminate set of data unavailable to demonstrate the influence on the rapidity and accuracy on earthquake parameters needed for tsunami warning.

Another remark is related to the reason of the stop of data due to the 2 recent hurricanes. Why data of so many stations where unavailable ? Power supply, destruction of station, transmission equipment, could be part of the response . . . It would be useful to

provide information on those issues. An additional point would be how to build robust stations to hurricane. Recommendations by the authors would be useful for all tsunami warning systems.

And last comment, the title is "Impact . . . on the PTWC Tsunami warning capability for the Caribbean Region " The tsunami warning system includes also sea-level stations . To be consistent with the title, the authors should add information on the availability of sea level data after the impact of these two recent hurricanes. These data are absolutely necessary to confirm whether a tsunami has been induced by the earthquake or not, and what are the characteristics of the tsunami waves (amplitude, period. . .). And provide some information how long it last to repair the stations. In case no impact was noticed on the availability of sea level data, the authors should shortly report on that.

Some minor corrections :

a) the notation 01:34 should be changed in . . . 1 mn 34 s or other format specifying minutes and seconds.

b) P5 l25 : Figure 10a = (Figure 6b + 110s) ; P5 l31 same correction

---

## Author Comment (AC1) · 29 Mar 2019

Thank you for taking the time to review the manuscript of write comments and suggestions to improve it. Please find our answers below.

1) The first comment is related to the magnitude referred. As related to fast assessment of earthquakes parameters, it is questionable to mention only the ML computation (page 5 line 11). It is well known that most of the Tsunami warning centers compute Mw, that is the more accurate magnitude in particular for last earthquakes and also typical "tsunami earthquake" events (see Kanamori 1975). Why the authors don't take into account the computation of Mw, and the variations of Mw accuracy depending on

[Figure]

the station available after the hurricanes?

Answer:

We do not consider our discussion of the ML magnitude method as questionable. We understand the interest in the Wphase magnitude method, but we consider a discussion of its accuracy as out of the scope and purpose of the paper. We will try to elucidate the reasons why while providing some background on the PTWC operational procedures. We cannot compute an earthquake's magnitude unless we first detect and locate its epicenter. When assessing the impact of the hurricanes on the PTWC operational capabilities for the region we applied the computation of the theoretical earthquake detection times as a way to quantify the impact of the hurricanes in a tangible, practical way. How much longer it takes to detect and locate an earthquake after the hurricanes turns into a direct expression of their actual operational impact.

The PTWC routinely locates and estimates the magnitude of global earthquakes with 5.7 or larger magnitudes. Messages issued for these events, however, always report the moment magnitude estimated via the P-wave moment magnitude, not the Wphase-based magnitude. Applying the Mwp method the PTWC can compute a quick estimate of an earthquake's magnitude in less than 5 minutes from origin time. Results from the Wphase method, however, take around 25 minutes, and although a regional implementation can reduce that time to 12~15 minutes from origin, these computations still take too long to consider them a replacement for the Mwp method, at least within the context of tsunami warning operations.

The available instrumental record of earthquakes in the Caribbean does not include many events with Mw magnitudes larger than 6.8. In its role as local tsunami warning center for Puerto Rico and the Virgin Islands (PRVI) the PTWC must routinely locate and compute the magnitude of earthquakes with 3.0 or larger magnitude. Moreover, in this role the PTWC must issue at least a Tsunami Information Statement (TIS) for any local earthquake in the vicinity of PRVI with a 4.0 or larger magnitude. Although the

[Figure]

ML magnitude saturates for earthquakes with magnitudes around 6.5, it still provides a fast magnitude estimate for the overwhelming majority of earthquakes occurring in the local vicinity of Puerto Rico and the Virgin Islands. For events with magnitudes larger than around 5.6 the PTWC will then compute first the Mwp moment magnitude, and later the Wphase-based magnitude, in that order. Once the estimated magnitude for an earthquake in the Caribbean reaches the 7.1 threshold the PTWC will issued a tsunami thread message. For earthquakes in the immediate vicinity of PRVI, however, the PTWC would issue a tsunami advisory message for shallow underwater earthquakes with a 6.5 or larger magnitude. As shown in our study, most of the impact from the hurricanes concentrated in the eastern Caribbean, so a discussion of the ML magnitude estimates turns paramount, as it constitutes the core of the PTWC local tsunami warning operations for Puerto Rico and the Virgin Islands.

In addition, the accuracy of magnitude estimations can vary, even for CMT magnitude estimates, by as much as 0.3 magnitude unit. Both the level of uncertainty inherent to the analysis methods currently applied, and the conservative criteria build into the PTWC operational procedures to cope with it make a discussion of Wphase magnitude estimates rather inconsequential within the scope of our study.

2) The second comment more general, the authors don't mention the reduction of the accuracy due to lack of data for the fast assessment of seismic parameters (location, depth, magnitude, etc...) To validate the results and conclusions of that study, a complementary study, using data set of recent large earthquakes in the region and eliminate the corresponding data (removing a set of data , i) the set corresponding to the stations stopped in consequence of the two 2017 hurricanes ; ii) other sets of data with several various hypothesis of path of future hurricane, southern, western Caribbean sea ...) would be the best demonstration, and quantify the impact of such weather disaster on the capacities of tsunami monitoring networks and warning systems.

Answer:

A characterization of the reduction of the accuracy due to lack of data for the fast assessment of the seismic parameters falls out of the scope of our study. The paper instead discusses the impact of the hurricanes on the most critical operational capability of a tsunami warning center, namely, its capacity to detect and locate earthquakes as fast as possible. Our study attempts to make something rather abstract more tangible by expressing the impact of the hurricanes as additional detection and response time delays. In our opinion, detection and response speeds turn not only easier to grasp than specific accuracy or error estimates, but also more critical for tsunami warning operations.

3) Another additional point related to fast seismic parameters computation and accuracy. The W-Phase centroid moment tensor computation is used at PTWC to get a fast tsunami threat forecast. The authors should provide the impact of data of large set of stations missing to the accuracy of the results of computation of W-Phase centroid moment tensor, considering only the set of stations available after the hurricanes. Similar complementary study could be performed considering one of the recent large earthquake in the region ( M > 7,0) and eliminate set of data unavailable to demonstrate the influence on the rapidity and accuracy on earthquake parameters needed for tsunami warning.

Answer:

The paper does not deal with the effect of the hurricanes on the Wphase centroid moment tensor computations or the PTWC tsunami forecasts. Please refer to our answers to comments 1) and 2) above.

4) Another remark is related to the reason of the stop of data due to the 2 recent hurricanes. Why data of so many stations where unavailable ? Power supply, destruction of station, transmission equipment, could be part of the response . . . It would be useful to provide information on those issues.

Answer:

Our colleagues from Puerto Rico and the Caribbean have reported the damage caused by the hurricanes to their seismic monitoring networks at different forums. Their accounts, combined with circumstantial evidence allows us to attribute the additional seismic station outages to the passing of both hurricanes, but we do not know the specific reasons for each particular site. In many cases the hurricanes destroyed the seismic station sites, in others their communications. It turns quite difficult to have an accurate record of what cause each particular seismic data outage. Despite the hurricanes specific effects in the field, those additional outages did affect the PTWC monitoring and warning capabilities. Due to this we adopted a pragmatic approach and used the available PTWC seismic data latency logs instead.

5) An additional point would be how to build robust stations to hurricane. Recommendations by the authors would be useful for all tsunami warning systems.

Answer:

We mention the need to build more robust seismic stations as part of the conclusions. The PTWC, however, does not install or maintain the seismic sites. We believe that the regional seismic operators in collaboration with the USGS and other organizations should draft these recommendations after conducting surveys of the actual damage caused by the hurricanes to each specific seismic site.

6) And last comment, the title is "Impact on the PTWC Tsunami warning capability for the Caribbean Region " The tsunami warning system includes also sea-level stations . To be consistent with the title, the authors should add information on the availability of sea level data after the impact of these two recent hurricanes. These data are absolutely necessary to confirm whether a tsunami has been induced by the earthquake or not, and what are the characteristics of the tsunami waves (amplitude, period ...). And provide some information how long it last to repair the stations. In case no impact was noticed on the availability of sea level data, the authors should shortly report on that.

Answer:

[Figure]

We agree that the water level data plays an important role when confirming the presence and actual size of a generated a tsunami. For tsunami warning purposes, however, water level data does not turn indispensable, particularly in the near field, except perhaps when dealing with sudden volcanic eruptions. To our knowledge, water level data monitoring and analysis has never prompted the issuance of a single tsunami warning or threat message. The PTWC issues tsunami warnings and threat messages based first and foremost of preliminary seismic data analysis, not water level data analysis. This has to do with both the detection speed possible with both types of data, as well as with the density of stations required for actual tsunami monitoring. Water level data provided by either the DART buoys or the coastal tide stations provides the means to confirm the presence of a tsunami, improve and adjust a tsunami forecast, and ultimately turn essential when deciding whether or not to issue a tsunami warning cancellation. As part of its operations, however, the PTWC geoscientists begin to actively monitor the water level stations closer to the earthquake's epicenter only after issuing a tsunami warning or threat message. Due to these reasons we opted for concentrating on the core of tsunami warning operations, and left the analysis of the water level data perhaps for another study.

7) Some minor corrections : a) the notation 01:34 should be changed in 1 mn 34 s or other format specifying minutes and seconds. b) P5 l25 : Figure 10a = (Figure 6b + 110s) ; P5 l31 same correction

Answer:

We will apply the suggested edits and suggestions.

---

## Referee Comment (RC2) · Ocal Necmioglu (Referee) · 30 Mar 2019

This paper discusses the considerable impact of the hurricanes in the Caribbean region to PTWC's seismic network which jeopardize PTWC's abilities in early determination of earthquake magnitude, especially essential in the issuance of the local tsunami warnings, and elaborates on remedy actions. Two parameters, data latency and azimuthal gap has been investigated in order to assess the degree of the loss in tsunami warning capabilities. Authors present detection delays of at least 15 seconds throughout at least 85% of the Caribbean reaching more than 30 seconds to affect 28% and 34% of the region for P and S wave detections, respectively, and azimuthal gaps starting from

180 degrees and exceeding even 270 degrees have been identified as the conditions of criticality. Authors argue that longer delays appear heavily concentrated within the area monitored by the PTWC in the eastern Caribbean.

As remedy actions, authors investigated the effect of i) reducing the number of P-phase picks required for event detection, and ii) reducing the number of S-phase picks required for preliminary ML magnitude computation, where i) results in a significant reduction of the P-wave detection times from a maximum additional delay of 163 seconds to the 66 seconds, and ii) results in reduction on the S-wave detection times from a maximum additional delay of 273 seconds to the 120 seconds. It is understood that still approximately 2 min delay in PTWC's response times should be considered despite the remedy actions considered. In their conclusions, the authors also suggest the network operators to consider to hurricane-proof at least a subset of their seismic stations, so as to maintain a minimum earthquake monitoring and local tsunami warning capability even if impacted by category 5 hurricanes when reinstalling damaged stations and rebuilding the supporting infrastructure.

The authors address an important challenge of an operational tsunami warning system in a multi-hazard context. A short discussion on the ML uncertainty as a result of reduced number of stations/phase readings, as ML is as a fast magnitude estimate suitable for the region complements also the detection and location of earthquakes, could support the valuable study provided by the authors. It would also be advisable to provide a bit more information on the reasons of the station availability (instrument damage, power outage, communication lines etc.) and average recovery times based on actual experiences, and preferably not only related to the seismic network but also sea-level network, if possible.

In their conclusion, the authors correctly point out that the devastating impact of hurricanes on the PTWC local tsunami warning capabilities at the local level highlights the vital, and potentially lifesaving role of educating the population to self-evacuate in the event of prolonged or strong ground shaking instead of waiting for official tsunami messages. It would be advisable to elaborate more on this important conclusion, due to the fact that this might be the one and only solution applicable to the local tsunami risk, even if the seismic networks perform in full. Caribe Wave tsunami exercises successfully conducted since 2011, for example, where the last one was conducted on 14 March 2019 with more than 830,000 participants, is probably the most important remedy action which could be referred to in this paper in addressing the technical/operational challenges of a local tsunami warning system to complete the end-to-end chain.

The Sendai Framework for Disaster Risk Reduction 2015-2030 recognizes the benefits of multi-hazard early warnings systems and places them in one of its seven global targets, namely to substantially increase the availability of and access to multi-hazard early warning systems and disaster risk information and assessments to people by 2030. Even though this falls out of the scope of this study, the authors in their conclusion may consider to provide a short elaboration on this aspect, specifically the need to analyse the feasibility and advantages of possible coupling the hurricane- and tsunami warnings in the multi-hazard context, especially in this region, to be addressed by another future publication perhaps. One should not forget the remaining big question: what happens if a tsunami occurs in this region during one of the peak moments of a hurricane? Even the means of self-evacuation may not exist anymore in such apocalypse scenario. . .

---

## Referee Comment (RC3) · Anonymous Referee #3 · 11 Apr 2019

Manuscript of Sardina et al. called "Impact of hurricanes Irma and Maria on the PTWC tsunami warning capability for the Caribbean region" represents rather a scientific report than a research paper where Authors theoretically evaluate PTWC response time capabilities before and after the two hurricanes. Authors first introduce their methodology to assess the expected PTWC event detection time, apply it to an ideal situation (all stations online and with zero data latency), then take into account the usual outage- and data latency statistics and, finally, consider network performance after the two devastating hurricanes. Their numerical analysis is extensively illustrated by a set of maps presenting event detection time as well as time delay introduced by the hurricanes.

[Figure]

The Manuscript is compact, clearly written, exemplifies an important question of TWC response time, and, to my opinion, should be published in NHESS after minor revisions.

In particular:

(1) From the text in Ch.3 is not clear if data latency and station outage statistics (Figure 1b, upper right corner, – note! – "right" not "left" as written in line 29 page 2) reflects the overall network performance during the second half of 2017 disregarding individual stations (i.e., data latencies and outages might "jump" from station to station within this time period), or latencies and outages are "bound" to particular stations? In the first case, results (detection time maps accounting for data availability) will strongly depend on how Authors distribute outage and latency statistics between concrete stations. In the second case (which, I think, is valid), it is not clear why such a statistics has a persistent character – why not to repair non-working stations (persistent outages)? Why not to reduce problematic data latency at correspondent stations?

(2) Ch. 5: One mitigation measure can be reduction of number of P-wave registering stations from 8 down to 4 (Figure 9a). How much should that affect the epicentral offset?

(3) At least for Figure 1 I would suggest to start the caption with: "Hypothetical epicentre positions coloured by theoretical detection time…..".

(4) Optional. Some figures could be send to Supplementary. For example, 2, 4, 5, 8, 10.

---

## Author Comment (AC2) · 12 Jun 2019

Thank you very much for your time and careful review of our paper. We truly appreciate your comments and suggestions. Please find below our answers to them.

C1: A short discussion on the ML uncertainty as a result of reduced number of stations/phase readings, as ML is as a fast magnitude estimate suitable for the region complements also the detection and location of earthquakes, could support the valuable study provided by the authors.

A1: Using fewer stations in the ML magnitude estimations would have an impact on

accuracy, but this gets somewhat mitigated by the fact that PTWC routinely computes ML magnitude corrections using the HUMP station as a reference. We did not addressed the accuracy of the ML magnitude estimates directly, however, because in practice it turns difficult to do, even under normal operational conditions. To verify the ML magnitude accuracy we would have to cross-validate them against the official catalog magnitudes, in this case those included in the NEIC catalog. The NEIC catalog, however, contains a variety of magnitudes for these rather small events, including mb and Md(duration) magnitudes, which makes a fair comparison difficult. Notwithstanding, we have done a comparison of the PTWC magnitude estimations for Caribbean events in the past, published as part of an article in Seismological Research Letters (SRL) in February of 2017 under the title "Evaluation of the Pacific Tsunami Warning Center's Performance for the Caribbean Based on the Compilation and Analysis of Tsunami Messages Issued between 2003 and July 2017". In general, PTWC magnitude estimations have a median residual of 0.2 magnitude unit when compared to the catalog magnitudes.

C2: It would also be advisable to provide a bit more information on the reasons of the station availability (instrument damage, power outage, communication lines etc.) and average recovery times based on actual experiences, and preferably not only related to the seismic network but also sea-level network, if possible.

A2: We do not know the specific reasons behind the outages at each seismic stations. We reported the end result on the PTWC end, regardless of the specific reasons. We know, however, that in many cases some stations suffered physical damage, while in other cases the communications' infrastructure collapsed due to the direct impact of the hurricanes. The Puerto Rico Seismic Network (PRSN), however, should have a database containing these specific data regarding the damages and the measures implemented in the aftermath of the hurricanes.

Regarding the network of see-level instruments, in this study we focused on the initial tsunami warning capability of the tsunami warning centers, which relies primarily on

the analysis of seismic data. The water level data turns very useful to confirm the presence of a tsunami, adjust tsunami forecasts, and ultimately to safely cancel a tsunami warning. The initial tsunami warning messages issued by the PTWC, however, rely first and foremost on the analysis of seismic data. To make our focus on this aspect of the PTWC operations more apparent, we will add the word "initial" to the title of our paper so that it reads "Impact of Hurricanes Irma and Maria on the PTWC Initial Tsunami Warning Capability for the Caribbean Region".

C3: In their conclusion, the authors correctly point out that the devastating impact of hurricanes on the PTWC local tsunami warning capabilities at the local level highlights the vital, and potentially lifesaving role of educating the population to self-evacuate in the event of prolonged or strong ground shaking instead of waiting for official tsunami messages. It would be advisable to elaborate more on this important conclusion, due to the fact that this might be the one and only solution applicable to the local tsunami risk, even if the seismic networks perform in full. Caribe Wave tsunami exercises successfully conducted since 2011, for example, where the last one was conducted on 14 March 2019 with more than 830,000 participants, is probably the most important remedy action which could be referred to in this paper in addressing the technical/operational challenges of a local tsunami warning system to complete the end-to-end chain.

A3: We agree with your assessment regarding the importance of both the role of educating the public to self-evacuate, and the Caribe Wave exercises as one of the most important remedy actions. We do not elaborate further or offer any specific recommendations, however, as a concrete course of action most come out as part of the conclusions of the working committees set up for these purposes. In a recent intervention during a workshop for disaster managers in Ponce, Puerto Rico, however, we mentioned the importance of these issues to those in attendance, mainly disaster managers from across Puerto Rico and the Virgin Islands.

C4: The Sendai Framework for Disaster Risk Reduction 2015-2030 recognizes the

benefits of multi-hazard early warnings systems and places them in one of its seven global targets, namely to substantially increase the availability of and access to multi-hazard early warning systems and disaster risk information and assessments to people by 2030. Even though this falls out of the scope of this study, the authors in their conclusion may consider to provide a short elaboration on this aspect, specifically the need to analyse the feasibility and advantages of possible coupling the hurricane- and tsunami warnings in the multi-hazard context, especially in this region, to be addressed by an other future publication perhaps. One should not forget the remaining big question: what happens if a tsunami occurs in this region during one of the peak moments of a hurricane? Even the means of self-evacuation may not exist anymore in such apocalypse scenario...

A4: We agree with the importance of these issues, which in themselves would require independent studies to address them appropriately. Indeed, the prospect of having to deal with a large tsunami while receiving a direct hit by a category 5 hurricane that wipes out most of the infrastructure seems like a worst case scenario. We mentioned these potential case scenarios to the disaster managers in attendance to the afore-mentioned workshop in Ponce, Puerto Rico.

---

## Author Comment (AC3) · 13 Jun 2019

Thank you for your careful review of our paper. We really appreciate your comments and suggestions. Please find below our answers to your comments and questions:

Q1: From the text in Ch.3 is not clear if data latency and station outage statistics (Figure 1b, upper right corner, – note! – "right" not "left" as written in line 29 page 2) reflects the overall network performance during the second half of 2017 disregarding individual stations (i.e., data latencies and outages might "jump" from station to station within this time period), or latencies and outages are "bound" to particular stations? In the first case, results (detection time maps accounting for data availability) will strongly depend

[Figure]

on how Authors distribute outage and latency statistics between concrete stations. In the second case (which, I think, is valid), it is not clear why such a statistics has a persistent character – why not to repair non-working stations (persistent outages)? Why not to reduce problematic data latency at correspondent stations?

A1: We will edit the text to correctly reference the figure, as indicated. The listed median data latencies reflect the overall performance of the available network, as those values in seconds reflect the most common latencies of the member stations. In other words, we use the median latencies as a representation of the most likely status of the network within a certain time interval. The latencies and outages at each station depend on several factors, mostly out of our control. In many cases the stations have communication issues and it takes time to repair them. This turns rather common for stations located in isolated islands, or hard to access sites. Repairing problematic stations depends on the resources available to each of the contributing seismic networks. The Puerto Rico Seismic Network (PRSN) has come a long way since the hurricanes, with most stations already online. We still have, however, a seismic data gap north of the Virgin Islands, possibly due to the difficulty of reaching remote stations in that area.

Q2: Ch. 5: One mitigation measure can be reduction of number of P-wave registering stations from 8 down to 4 (Figure 9a). How much should that affect the epicentral offset?

A2: We reduced to 4 the number of P-picks as a temporary measure due to the lack of enough seismic stations to detect earthquakes with the require speed in the aftermath of the hurricanes. Reducing to 4 the number of P-picks required for a preliminary location aimed at releasing the preliminary locations for manual review by the analyst on duty faster than otherwise possible. Manual review would then increase the number of P-picks whenever possible. In this regard, however, the L-shape topology of the network, which leads to azimuth gaps of more than 180 degrees for a large number of events in the area seems to eclipse the number of P-picks as the main factor affecting the accuracy of the PTWC preliminary locations. We published some

statistics regarding the accuracy of the PTWC preliminary earthquake parameters for Caribbean earthquakes as part of an article in Seismological Research Letters (SRL) in February of 2017 under the title "Evaluation of the Pacific Tsunami Warning Center's Performance for the Caribbean Based on the Compilation and Analysis of Tsunami Messages Issued between 2003 and July 2017". In addition to this study, further compilation of PTWC earthquake preliminary parameters and their cross-validation with the solutions listed in the NEIC catalog indicate that the median epicentral offset went from 11.8 km during 2016 to 16.2 km during 2017, which we could associate with the reduce number of stations due to the impact of both hurricanes. The median epicentral went down to 11.3 km during 2018, and 11.5 km up to April, 2019.

Q3: At least for Figure 1 I would suggest to start the caption with: "Hypothetical epicentre positions coloured by theoretical detection time A3: Indeed, the suggestion turns more technically accurate, but possibly more difficult to understand for people with less knowledge of the subject. Notwithstanding, we will consider the suggestion before submitting the final version of the paper.

Q4: Optional. Some figures could be send to Supplementary. For example, 2, 4, 5, 8, 10.

A4: In our opinion, leaving those figures out of the main text would make it harder to understand the results and follow along the narrative of the paper. For this reason, we ended up including them all, so that people can see where subsequent maps came from, and how we attained them.

---

## Author Response (AR1)

List of changes, also highlighted in red in the revised manuscript:

1) Added the word "Initial" to the title of the paper.

2) Page 1, Line 3: Added the word "initial" to the abstract to make it consistent with the slightly modified title.

3) Page 2, Line 3: Added the sentence "The PTWC issues its initial tsunami messages relying entirely on the analysis of near-real time seismic data provided by the available seismic networks, not water level data."

4) Page 2, Line 4: Replaced "We" with "Consequently we" to account for the inclusion of the new sentence in 3) above.

5) Page 3, Line 1: Replaced "left", with "right" as correctly pointed out by one of the reviewers.

6) Page 10, Figure 1 label: Added "Hypothetical epicenter positions coloured by the" as suggested by one of the reviewers.